# Disease progression of 213 patients hospitalized with Covid-19 in the Czech Republic in March–October 2020: An exploratory analysis

Martin Modrák[1]*, Paul-Christian Bürkner[2], Tomáš Sieger[3], Tomáš Slisz[4,5], Martina Vašáková[4,5], Grigorij Mesežnikov[6], Luis Fernando Casas-Mendez[6], Jaromír Vajter[6], Jan Táborský[7], Viktor Kubricht[8], Daniel Suk[9], Jan Horejsek[9], Martin Jedlička[10], Adriana Mifková[10], Adam Jaroš[11], Miroslav Kubiska[12], Jana Váchalová[12], Robin Šín[12], Markéta Veverková[13], Zbyšek Pospíšil[14], Julie Vohryzková[15], Rebeka Pokrievková[16], Kristián Hrušák[15], Kristína Christozova[15], Vianey Leos-Barajas[17], Karel Fišer[18], Tomáš Hyánek[11]

1 Bioinformatics Core Facility, Institute of Microbiology of the Czech Academy of Sciences, Prague, Czech Republic, 2 Cluster of Excellence SimTech, University of Stuttgart, Stuttgart, Deutschland, Germany, 3 Dept. of Cybernetics, Faculty of Electrical Engineering, Czech Technical University in Prague, Prague, Czech Republic, 4 Department of Respiratory Medicine, 1st Faculty of Medicine, Charles University in Prague, Prague, Czech Republic, 5 Thomayer University Hospital, Prague, Czech Republic, 6 Motol University Hospital, Prague, Czech Republic, 7 AGEL Hospital Nový Jičín, Nový Jičín, Czech Republic, 8 Kralovské Vinohrady University Hospital, Prague, Czech Republic, 9 General University Hospital in Prague, Prague, Czech Republic, 10 Military Hospital Olomouc, Olomouc, Czech Republic, 11 Na Homolce Hospital, Prague, Czech Republic, 12 Department of Infectious Diseases and Travel Medicine, Faculty of Medicine in Pilsen, Charles University, University Hospital in Pilsen, Pilsen, Czech Republic, 13 Hořovice Hospital, Hořovice, Czech Republic, 14 Třebíč Hospital, Třebíč, Czech Republic, 15 2nd Faculty of Medicine, Charles University in Prague, Prague, Czech Republic, 16 3rd Faculty of Medicine, Charles University in Prague, Prague, Czech Republic, 17 Department of Statistical Sciences, University of Toronto, Toronto, Canada, 18 Department of Bioinformatics, 2nd Faculty of Medicine, Charles University in Prague, Prague, Czech Republic

* martin.modrak@biomed.cas.cz

**Data Availability Statement:** The data that support the findings of this study are available on request from the corresponding author or the secretariat of

## Abstract

We collected a multi-centric retrospective dataset of patients (N = 213) who were admitted to ten hospitals in Czech Republic and tested positive for SARS-CoV-2 during the early phases of the pandemic in March—October 2020. The dataset contains baseline patient characteristics, breathing support required, pharmacological treatment received and multiple markers on daily resolution. Patients in the dataset were treated with hydroxychloroquine (N = 108), azithromycin (N = 72), favipiravir (N = 9), convalescent plasma (N = 7), dexamethasone (N = 4) and remdesivir (N = 3), often in combination. To explore association between treatments and patient outcomes we performed multiverse analysis, observing how the conclusions change between defensible choices of statistical model, predictors included in the model and other analytical degrees of freedom. Weak evidence to constrain the potential efficacy of azithromycin and favipiravir can be extracted from the data. Additionally, we performed external validation of several proposed prognostic models for Covid-19 severity showing that they mostly perform unsatisfactorily on our dataset.

the Institute of Microbiology of the Czech Academy of Sciences (contact via mbu@biomed.cas.cz) for researchers who meet the criteria for access to confidential data. The data are not publicly available due to privacy restrictions imposed by the Ethical committee of General University Hospital in Prague and the GDPR regulation of the European Union. We will be happy to arrange to run any analytical code locally and share the results, provided the code and the results do not leak personal information.

**Funding:** MM was supported by ELIXIR CZ research infrastructure project (Ministry of Youth, Education and Sports of the Czech Republic, Grant No: LM2018131, https://www.msmt.cz/) including access to computing and storage facilities. The funders had no role in study design, data collection and analysis, decision to publish, or preparation of the manuscript.

**Competing interests:** The authors have declared that no competing interests exist.

## Introduction

The Covid-19 pandemic caused by severe acute respiratory syndrome coronavirus (SARS-CoV-2) has, as of June 2021, led to over 172 million cases and over 3.7 million deaths. The present study was designed and conducted during March—October 2020, when Czech Republic experienced a relatively mild first wave of the pandemic due to early and strict lockdowns. Low numbers of cases continued throughout the summer but during September and October, after most of the data collection for this study concluded, the number of cases was raising again. On October 1st 2020, Czech Republic had accumulated 74283 total confirmed Covid-19 cases and 704 confirmed Covid-19 related deaths [1].

At the time the study was conducted, the proposed treatments included antivirals approved for other indications (chloroquine, hydroxychloroquine, lopinavir/ritonavir, remdesivir, favipiravir, umifenovir), azithromycin, corticosteroids, immunoglobulins, tocilizumab and convalescent plasma [2, 3]. Notably, the anti-malarial and anti-rheumatic drug hydroxychloroquine and the macrolide antibiotic azithromycin showed promise in early data and were broadly available and thus were frequently used in the early stages of the pandemic. Remdesivir, previously designed and approved for Ebola, SARS and MERS, also reported good initial results. However, during spring and summer 2020 remdesivir was available in Czech Republic only in limited amounts via compassionate use programme. The RECOVERY trial reported positive results of coroticosteriod dexamethasone for severe cases in June 2020 [4], but at this point the number of Covid-19 patients hospitalized in Czech Republic was low and dexamethasone thus did not see wider use until later in the pandemic.

Our understanding of the efficacy of Covid-19 treatments has improved substantially since the present study was conducted. As of April 2021, the pharmacological treatments that were deemed to be beneficial for at least one outcome in a systematic review of randomized trials were the corticosteroid dexamethasone (mortality, mechanical ventilation), colchicine (mortality, length of hospital stay), the antiviral remdesivir (mechanical ventilation), Janus kinase inhibitors (mechanical ventilation, duration of ventilation), IL-6 inhibitors (mechanical ventilation, length of hospital stay), the antiviral favipiravir (length of hospital stay, resolution of symptoms) and the anti-androgen proxalutimide (admission to hospital). Hydroxychloroquine, interferon beta, lopinavir-ritonavir, azithromycin, vitamin C, vitamin D, anticoagulants and ACE inhibitors were considered to not be better than standard of care and lopinvair-ritonavir showed evidence of harm, although most of the conclusions were considered to be of low certainty [5]. Interestingly, in observational studies, hydroxychloroquine was often found to be associated with better outcomes [6–8]. No benefit was also observed in a meta-analysis of randomized trials of convalescent plasma treatment [9].

High IL-6, D-dimer values were observed to be associated with worse outcome and increased disease severity [10]. Large study of electronic health records [11] showed an increase in C-reactive protein in early disease and increase of D-dimer and white blood cell count in later stages of the disease.

An ongoing challenge in evaluating Covid-19 treatments is that the analysis and interpretation of the data is often inappropriate or misleading, most notably interpreting lack of evidence due to small sample size as evidence of no effect [12, 13].

Additionally, many methods for predicting disease severity of Covid-19 were published, but the methods are at high risk of bias and lack external validation [14].

The present study aims to describe the outcomes and disease course of hospitalized patients with mild to severe clinical presentation in a multicentric Czech cohort during the early stages of the pandemic, explore the association between the outcomes and pharmacological

interventions and to provide external validation to previously published prognostic models for Covid-19 severity.

## Methods

### Patients and study design

A convenience sample of patients from 10 sites was collected. The study sites span the whole spectrum of sizes from large university hospitals in major cities with multiple dedicated Covid-19 wards (Thomayer University Hospital in Prague, Motol University Hospital in Prague, Kralovské Vinohrady University Hospital in Prague, General University Hospital in Prague, University Hospital in Pilsen) through major regional/specialized hospitals (Na Homolce Hospital in Prague, Military Hospital Olomouc) as well as smaller hospitals caring for just several Covid-19 patients at a time (AGEL Hospital Nový Jičín, Hořovice Hospital, Třebíč Hospital). The sites were chosen based on availability and willingness of the personnel to participate in data collection. None of the study sites was exclusively dedicated to treating Covid-19 patients. For each site, the dataset contains all patients hospitalized in the participating wards over the data collection period. The data collection started at the onset of the Covid-19 pandemic in March 2020 (except for one site where some older records were inaccessible), but the end date for collection differs between sites due to time constraints of the participating physicians. Three sites included total of 23 patients that could be considered part of "second wave" (admitted after September 1st). Last patient included in the dataset was admitted on October 12th. See S1 Fig for per-site data collection periods. Patients over the age of 18 were included if they had PCR-confirmed infection of SARS-CoV-2 and were not participating in a clinical trial of any Covid-19 pharmacotherapy.

Not all patients developed pneumonia or other symptoms of Covid-19. All patients received the standard of care which could include supplemental oxygen and ventilation and antibiotics for bacterial superinfections, as determined by the attending physician. Some patients were not indicated for all treatment modalities (especially mechanical ventilation) based on decision of the attending physician and underlying patient condition. We note that the participating sites were not homogeneous in either patient population or treatment protocols. The choice of pharmacological treatment was based on the decision of the attending clinician and its availability.

The study was approved by the Ethical committees of General University Hospital, Hospital Nový Jíčín, Motol University Hospital, Thomayer Hospital, University Hospital Vinohrady, Military Hospital Olomouc, Na Homolce Hospital, University Hospital in Pilsen, Hořovice Hospital, Jihlava Hospital, all data were collected in fully anonymized form. Data was collected between June and October 2020 for patients that were treated between March and October 2020.

### Data collection

We collected data on comorbidities and information about disease progression on daily resolution including breathing support required, oxygen flow rate, experimental anti-Covid-19 and antimicrobial drugs taken and several laboratory markers (PCR positivity for SARS-CoV-2, C-reactive protein, D-dimer, Interleukin 6, Ferritin, lymphocyte count). Full protocol for data collection is attached in S1 File and the data collection table in S2 File. Due to very low number of patients using extra-corporeal membrane oxygenation (N = 1) or non-invasive positive pressure ventilation (N = 6) in our sample, we merged those categories with mechanical ventilation.

## Statistical analysis

The character of the convenience sample does not allow for a proper assessment of the association between treatments and patient outcomes, because the treatments had not been assigned to patients at random but were only observed retrospectively. This can be partially remedied by adjusting for patient characteristics in the analysis, but such adjustments will always be imperfect and the analysis needs to be treated as exploratory and interpreted cautiously.

Since many details of analysis may influence the conclusions made, we performed multiverse analysis [15] and report results for all the hypothesis tested across multiple different models using both frequentist and Bayesian paradigms. For each model class we worked with several possible sets of adjustments. All analyses were performed in the R language [16], visualization and data cleaning was run with the `tidyverse` package [17].

First class of models are frequentist survival and multistate models under the proportional hazards assumption as implemented in the `coxph` function from the `survival` package [18]. We primarily use a model with competing risks for death and discharge from hospital (see Fig 1a).

Second class of models are Bayesian hidden Markov models (HMM) of disease progression implemented via a custom extension to the `brms` package [19]. The parametrization of the HMM is inspired by Williams et al. [20]: the actual process of disease is assumed to be continuous and allow only for transitions between neighboring states (as shown in Fig 1b and 1c). The total probability of transition between any two states over the period of a day is then computed as the total probability of transition across all possible paths. This class of models does not satisfy the proportional hazards assumption, instead, it is assumed the process has the Markov property—i.e., that the (potentially unobserved) state and the covariates at a given day contain all the information to determine probabilities of the states on the next day. We use two versions of such models, one working solely with the observed breathing support and one assuming a hidden improving/worsening distinction. All of the hidden Markov models take into account whether best supportive care was initiated and a patient was thus not indicated to progress to more intensive treatment modalities.

Finally, we used a set of Bayesian regression models implemented with the `brms` package [19]. Those included overall survival, state at day 7 or 28 as either binary or categorical outcome and a Bayesian version of the Cox proportional-hazards model.

Except for age, sex and comorbidities, all covariates are treated as time-varying, e.g., the effect of taking a drug is only included for the days after the drug was taken. More details on the exact model formulations can be found in the supplementary statistical analysis in S3 File.

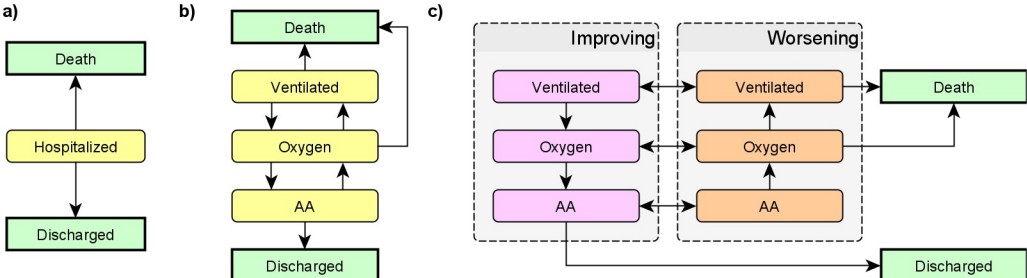

**Fig 1. States used in the competing risk model (a) and in the two hidden Markov model variants (b,c).** AA = Ambient air, Oxygen = Nasal oxygen, Ventilated = any form of ventilation (non-invasive positive-pressure ventilation, mechanical ventilation and extra-corporeal membrane oxygenation). In all models the 'Death' and 'Discharged' states are terminal. In the second hidden Markov model (c), the 'Improving' and 'Worsening' variants of each non-terminal state are not observable—only the breathing support is observed and improving/worsening is inferred from progression of the disease.

## Evaluating prognostic models

We searched the living systematic review of Covid-19 prognostic models [14] for those that could be applied to our dataset (i.e., where we have gathered all the input features). We primarily focused on the Area Under Receiver Operating Characteristic Curve (AUC), and its bootstrapped 95% confidence intervals which we computed using the `pROC` package [21]. When there were multiple reasonable ways to evaluate the outcome or a predictor in our dataset, we computed and reported all of those options. We used two simple scores with age or the decade of age as the sole predictor to have a baseline to compare the scores against.

Complete code for all analyses is available at https://github.com/cas-bioinf/covid19retrospective/.

## Results

### Baseline characteristics

In total, we were able to gather data for 213 patients, see Table 1 for the overall characteristics of the patient sample and several subgroups we used in the analysis, including treatments taken. Counts of all treatment combinations are shown in S2 and S3 Figs shows outcomes by

**Table 1. Patient characteristics for the overall sample and treatment subgroups.** Note that the favipiravir subgroup is not exclusive with either the HCQ or No HCQ group.

|  | All | HCQ | No HCQ | Favipiravir |
|---|---|---|---|---|
| N | 213 | 108 | 105 | 9 |
| Distinct sites | 10 | 10 | 10 | 1 |
| Male | 105 (49%) | 53 (49%) | 52 (50%) | 4 (44%) |
| Age (mean, IQR) | 69 (58–80) | 67 (56–80) | 70 (64–82) | 59 (51–68) |
| Admitted for Covid | 172 (81%) | 96 (89%) | 76 (72%) | 8 (89%) |
| Took hydroxychloroquine | 108 (51%) | 108 (100%) | 0 (0%) | 8 (89%) |
| Took azithromycin | 72 (34%) | 63 (58%) | 9 (9%) | 8 (89%) |
| Took dexamethasone | 4 (2%) | 0 (0%) | 4 (4%) | 0 (0%) |
| Took favipiravir | 9 (4%) | 8 (7%) | 1 (1%) | 9 (100%) |
| Took remdesivir | 3 (1%) | 0 (0%) | 3 (3%) | 0 (0%) |
| Convalescent plasma | 7 (3%) | 6 (6%) | 1 (1%) | 3 (33%) |
| Ischemic Heart Disease | 43 (20%) | 15 (14%) | 28 (27%) | 1 (11%) |
| Takes antihypertensives | 141 (66%) | 66 (61%) | 75 (71%) | 4 (44%) |
| Heart Failure | 34 (16%) | 15 (14%) | 19 (18%) | 0 (0%) |
| COPD | 21 (10%) | 9 (8%) | 12 (11%) | 1 (11%) |
| Asthma | 18 (8%) | 9 (8%) | 9 (9%) | 0 (0%) |
| Other lung disease | 14 (7%) | 7 (6%) | 7 (7%) | 0 (0%) |
| Diabetes | 51 (24%) | 18 (17%) | 33 (31%) | 3 (33%) |
| Renal Disease | 43 (20%) | 19 (18%) | 24 (23%) | 2 (22%) |
| Liver Disease | 14 (7%) | 8 (7%) | 6 (6%) | 1 (11%) |
| Smoking | 28 (13%) | 15 (14%) | 13 (12%) | 1 (11%) |
| BMI (mean, IQR) | 28 (24–31) | 28 (24–30) | 28 (24–31) | 32 (27–35) |
| Best supportive care | 58 (27%) | 22 (20%) | 36 (34%) | 0 (0%) |
| Death | 42 (20%) | 15 (14%) | 27 (26%) | 0 (0%) |
| Discharged | 122 (57%) | 76 (70%) | 46 (44%) | 8 (89%) |

IQR = interquartile range, COPD = Chronic obstructive pulmonary disease, BMI = body-mass index, Best supportive care = patient was not indicated to undergo more intensive treatment modality.

study site, demonstrating quite large hospital-specific differences. The dataset includes 19 patients already reported in a study of inflammatory signatures of Covid-19 [22].

## Disease progression

In Fig 2 we show the overall disease progression for all patients and in Fig 3 we show the time-course of a subset of the markers we have measured. The data show some interesting patterns: patients with low Interleukin-6 or D-dimer values are overrepresented among patients with better outcomes, most patients had high CRP upon admission and for many the CRP levels stayed elevated over the whole hospitalization. However, the limited nature of the data does not allow for any statistically robust conclusions. We also see that the marker levels were not substantially stratified by study site. Those patterns should however be interpreted with care due to systematic missingness of the data—in particular, patients that fared worse were probably more likely to have the markers measured. However, we believe this kind of patient-level view is useful to appreciate the extent of both between-patient and within-patient variability.

The between-patient variability is notable even across outcomes—when ordering the patients by the highest CRP levels experienced throughout the hospital stay, the top 20% of patients that breathed ambient air for the whole hospitalization experienced higher levels than the bottom 20% of patients that required ventilation or died. This overlap is even larger when comparing only against the patients that died and D-dimer, Interleukin-6 and lymphocyte count also show a notably larger overlap than CRP (S4 Fig).

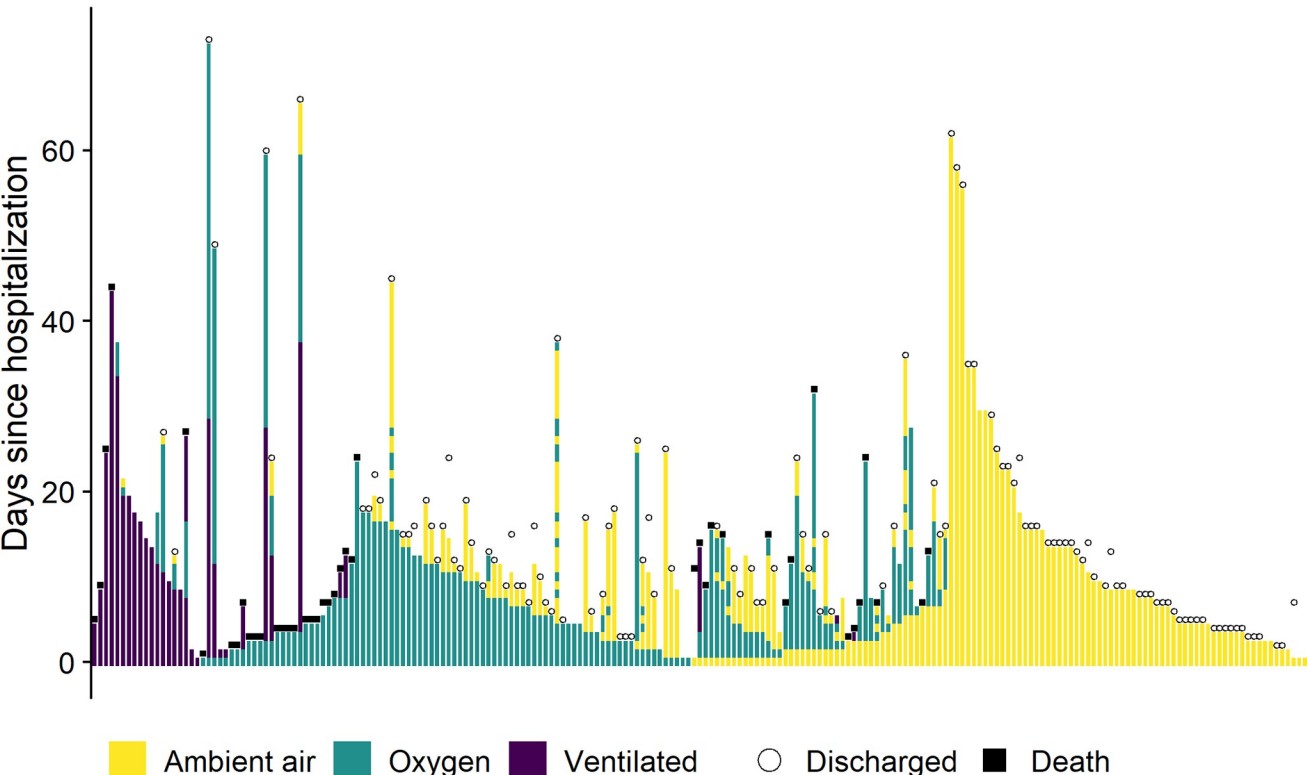

**Fig 2. Disease progression for all patients included in the study as determined by breathing support required.** Each vertical strip is a single patient, the ordering on the horizontal axis is by disease severity. Ventilated = any form of ventilation (non-invasive positive-pressure ventilation, mechanical ventilation and extra-corporeal membrane oxygenation).

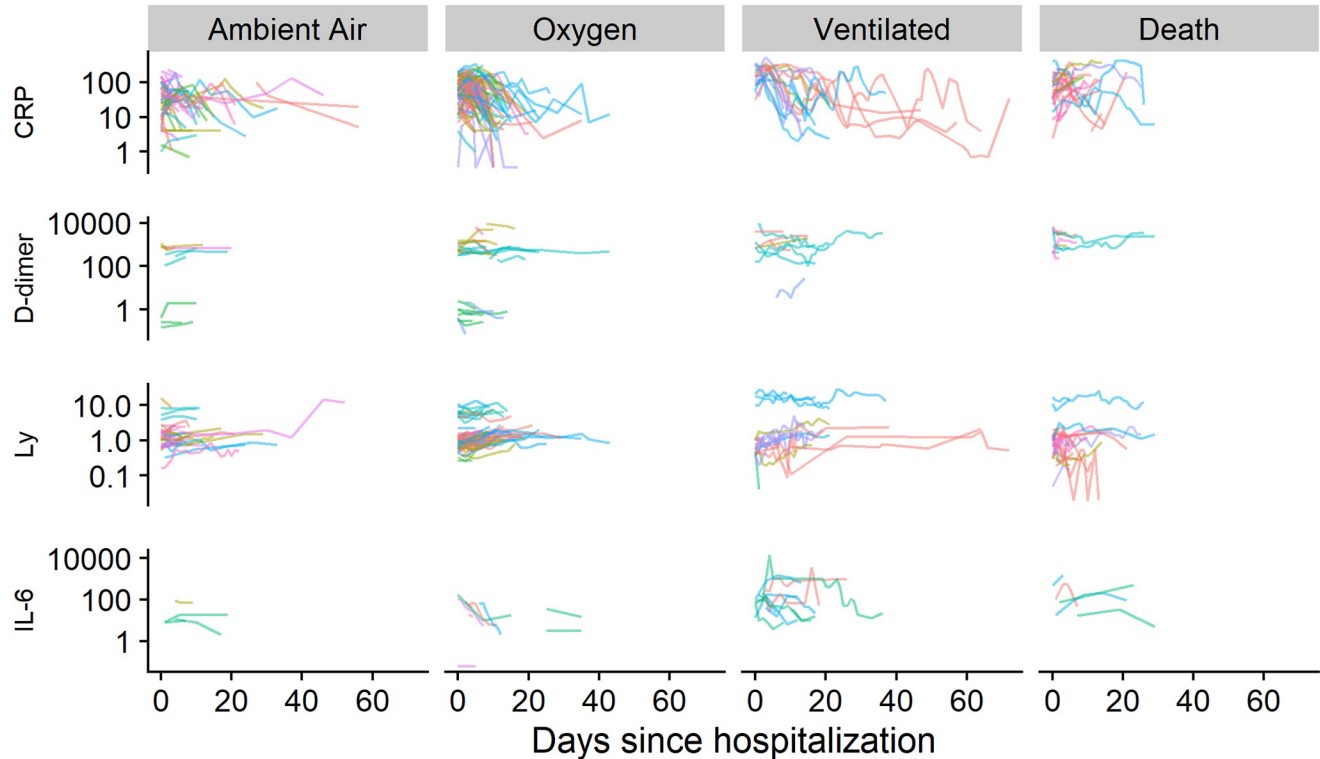

**Fig 3. Values of selected markers over the course of the disease.** Each line represents a patient, stratified by the worst breathing support required over the hospitalization. Color indicates study sites. The vertical scale is logarithmic. Ventilated = any form of ventilation (non-invasive positive-pressure ventilation, mechanical ventilation and extra-corporeal membrane oxygenation), CRP = C-reactive protein [mg/l], D-dimer [ng/ml DDU], Ly = lymphocyte count [$10^9$/l], IL-6 = Interleukin 6 [ng/l].

## Association between patients' characteristics and treatments

As noted above, the nature of the convenience sample did not enforce random assignment of treatments to patients. In fact, patients with worse baseline characteristics, which lead to worse outcomes, were less likely to receive hydroxychloroquine (see Fig 4). This clearly creates a bias towards a positive effect of hydroxychloroquine on the outcome (and potentially for other treatments as well—most were used in combination with hydroxychloroquine), which, how-ever, could be false.

Taken quantitatively, the comorbidities known upon hospitalization were informative with respect to the future hydroxychloroquine treatment: the score representing the cumulative presence of ischemic heart disease, hypertension drugs, former heart failure, COPD, other lung diseases, renal disease, or high creatinine was associated with a lower chance of taking hydroxychloroquine over the course of the hospitalization (the chance was only 79.9%, 95% confidence interval (65.3, 97)%, Chi-square test in the logistic regression model, $\chi^2 = 5.18$, df = 1, P = 0.023).

## Association between treatments and outcomes

Here, we focus on hydroxychloroquine and azithromycin as those are the only treatments with larger sample size. We also investigate favipiravir as it is less well reported in the literature. Hydroxychloroquine was dosed almost exclusively in a 5-day regime starting with a loading dose of 800mg on the first day and followed by 400mg. Majority of patients complemented

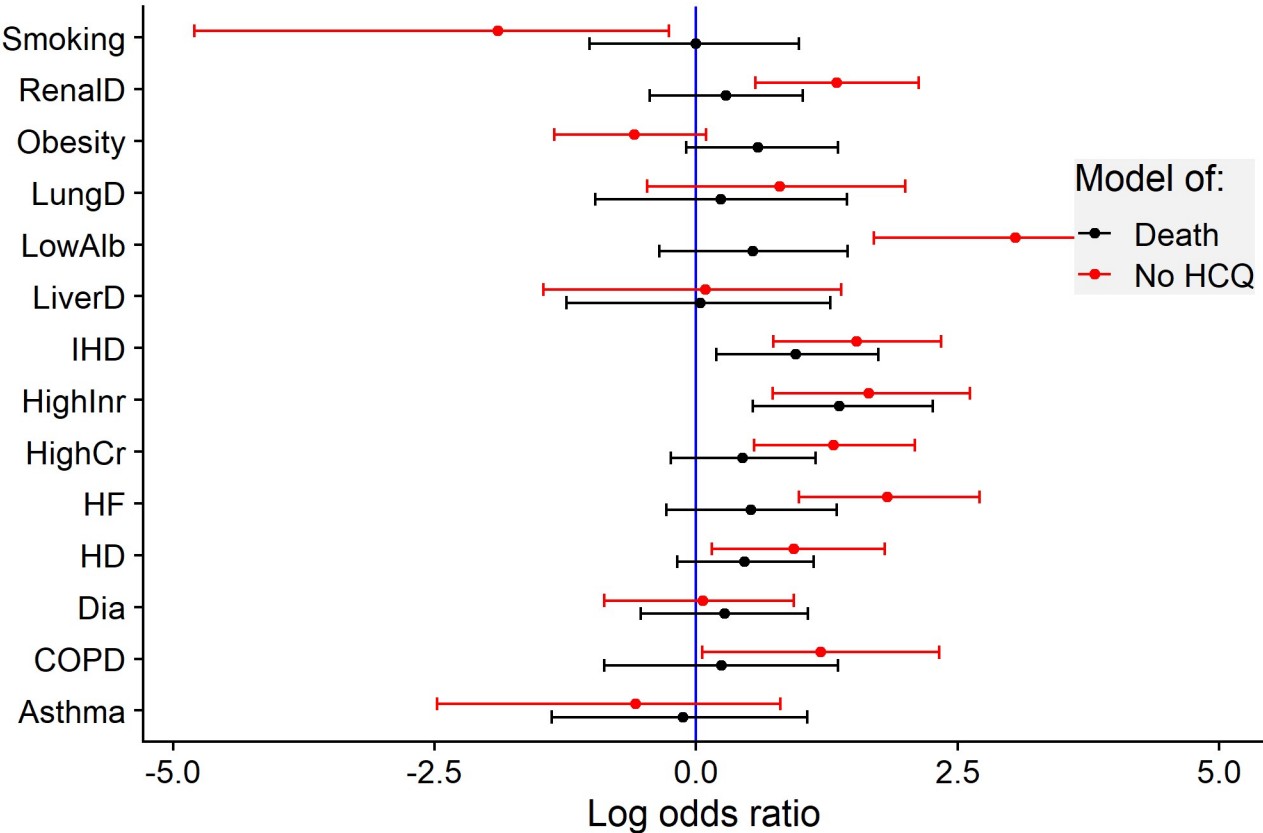

**Fig 4. The choice of treatment with hydroxychloroquine seemed to be associated with the status of patients upon hospitalization.** Comorbidities were associated with both worse outcome (black) and lower chance of treatment with hydroxychloroquine (red). Dots and lines represent the estimates and the 95% confidence intervals of the log odds ratio of the respective outcome. HCQ: hydroxychloroquine, IHD: ischemic heart disease, HD: hypertension drugs, HF: heart failure history, COPD: chronic obstructive pulmonary disease, LungD: other lung disease, Dia: diabetes, RenalD: renal disease, LiverD: liver disease, HighCr: creatinin above 115 for males or above 97 for females, HighInr: Prothrombin time (Quick test) as International Normalized Ratio above 1.2, LowAlb: albumin in serum/plasma below 36 g/l.

hydroxychloroquine with azithromycin while azithromycin was rarely used alone (see Table 1). Azithromycin was most frequently dosed 250 or 500mg/day, but doses ranging from 100mg/day to 1500 mg/day were observed. Favipiravir was used only at one site with a loading dose of 3600mg on the first day, followed by at most 9 days with a 1600mg dose. All but one of the patients receiving favipiravir also received hydroxychloroquine. Treatment was initiated mostly within two days of admission (see S5 Fig).

The results of the multiverse analysis for association between hydroxychloroquine, azithromycin and favipiravir usage and death is shown in Fig 5—here, we only show models that were not found to have immediate problems representing the data well or computational issues (see S3 File for details). Results for all models we tested are reported in S6–S8 Figs, with additional details in S3 File. The results do not change noticeably when only patients from the first wave are included (S6–S8 Figs).

Most models report that using hydroxychloroquine is associated with lower risk of death. We must however bear in mind the potential bias noted in the previous section. Also, we see that for the HMM models, as we add adjustments the credible intervals do not widen but instead shift towards zero. This is a weak indication that further adjustments could drive the effect towards zero. We did not attempt to model additional adjustments as the models became

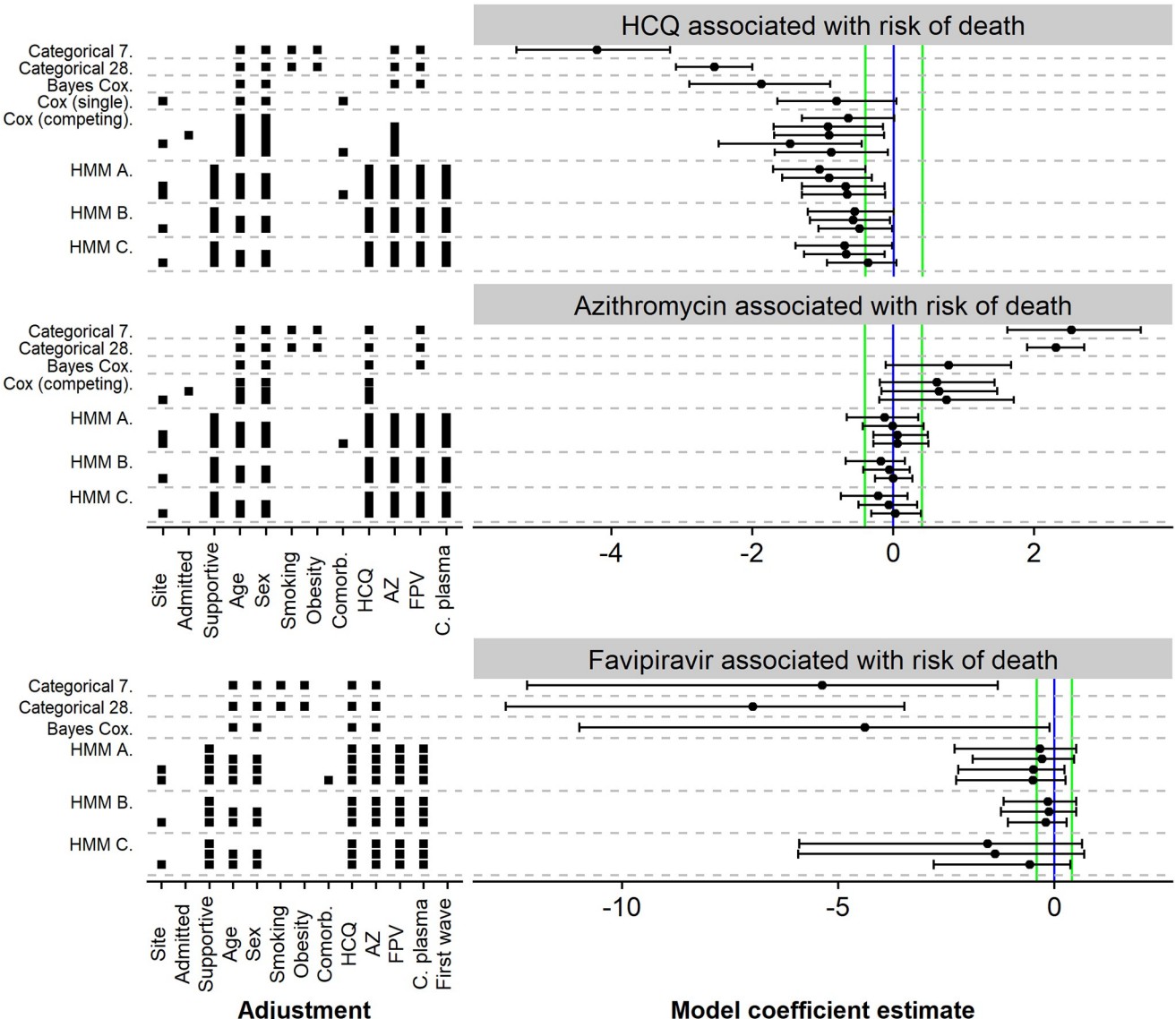

**Fig 5. Estimates of model coefficients for association between treatments and main outcomes.** Each row represents a model—Categorical 7/28 = Bayesian categorical regression for state at day 7/28, Bayes Cox = Bayesian version of the Cox proportional hazards model with a binary outcome, Cox (single) = frequentist Cox model with a binary outcome, Cox (competing) = frequentist Cox model using competing risks (as in Fig 1a), HMM A = Bayesian hidden-Markov model as in Fig 1b with predictors for rate groups, HMM B = Bayesian hidden-markov model as in Fig 1b with predictors for individual rates, HMM C = Bayesian hidden-Markov model as in Fig 1c. For frequentist models, we show maximum likelihood estimate and 95% confidence interval, for Bayesian models we show posterior mean and 95% credible interval. The estimands are either log odds-ratio (Categorical, HMM) or log hazard ratio (Cox variants). In all cases coefficient <0 means better patient outcome in the treatment group. Vertical lines indicate zero (blue) and substantial increase or decrease with odds or hazard ratio of 3:2 or 2:3 (green). Additionally the factors the model adjusted for are listed—Site = the study site, admitted = Admitted for Covid-19, Supportive = best supportive care initiated, Comorb. = total number of comorbidities, AZ = took azithromycin, HCQ = took hydroxychloroquine, FPV = took favipiravir, C. plasma = received convalescent plasma.

computationally unstable. The case of hydroxychloroquine serves as a "control group" for our other results—since randomized trials give us high confidence that hydroxychloroquine does not substantially reduce mortality, we can be quite certain the associations we observe for hydroxychloroquine are just a measure of bias in the data. Additionally, our models either cannot determine the sign of association between azithromycin and risk of death or even show an

increase in risk of death. This serves as a weak evidence that a substantial improvement in mortality from azithromycin is unlikely.

Most models exclude very strong association between increased risk of death and using favipiravir, but our data are necessarily quite limited, which is reflected in the very wide uncertainties around estimates. We also cannot put strict bounds on the association between favipiravir and length of hospitalization.

We also examined the association between treatments and length of hospital stay for all the patients that survived. Almost all models cannot discern the sign of the association for all treatments examined (S6–S8 Figs). Similarly, we studied the association between D-dimer and Interleukin 6 and outcomes, with unconclusive results as well (S9 Fig).

## Published prognostic models are not better than using age as the sole predictor of outcome

Following Wynants et al. [14] we found five prediction models we were able to recompute: Li et al. report the ACP index [23] combining CRP and age to form 3 grades, Chen & Liu [24] report a continuous score using age, CRP, D-dimer and lymphocyte count, Shi et al. [25] use age, sex and hypertension to form 4 grades, Caramelo et al. use age, sex, hypertension, diabetes, cardiac disease, chronic respiratory disease and cancer to form a continuous score [26], Bello-Chavolla et al. [27] use age, diabetes, obesity, pneumonia, chronic kidney disease, COPD and immunosuppression to build a score ranging from -6 to 22. For the latter two scores we had to impute some of the predictors as they had no immediate equivalent in our dataset. The outcomes present in the studies were: 12-day mortality, 30-day mortality and mortality without any further details, here we report results for both 12-day and 30-day mortality. Full details on the scores and how we used our dataset to compute them is given in the S3 File.

All prognostic models we tested performed similarly to or notably worse than using age as the only predictor and also worse than originally reported (Fig 6). Additionally, some publications did not provide enough detail to unambiguously reconstruct how the score and/or outcome was assessed. We thus concur with Wynants et al. [14] that reported prediction scores are at high risk of bias and need additional careful evaluation.

## Discussion

Our data show the extent of between-patient variability in progression of the disease in terms of both length of hospital stay, duration of various types of breathing support and basic markers. A direct comparison with other studies is hard to perform as almost always only summaries of measurements are reported.

For multiple candidate Covid-19 treatments, observational data repeatedly contradicted results of randomized controlled trials (contrast e.g. Catteau et al. [6] to the RECOVERY trial [28] for hydroxychloroquine and Liu et al. [29] to Agarwal et al. [30] for convalescent plasma). Our results for hydroxychloroquine also fit into this pattern. This should make us wary about over-interpreting the results of this study for azithromycin and favipiravir, although some higher-quality evidence that suggests clinical benefit of favipiravir has been reported [5].

The current (April 2021) Covid-19 treatment guidelines in Czech Republic recommend monoclonal antibodies and in some cases convalescent plasma or favipiravir as early treatment for high-risk patients with mild or no symptoms. For more severe cases dexamethason and anticoagulants are recommended while remdesivir is recommended only for patients that have severe disease but do not require mechanical ventilation [31]. This is similar to recommendations from the National Institute of Health in the USA who additionally recommend tocilizumab in some cases while not recommending convalescent plasma and favipiravir [32].

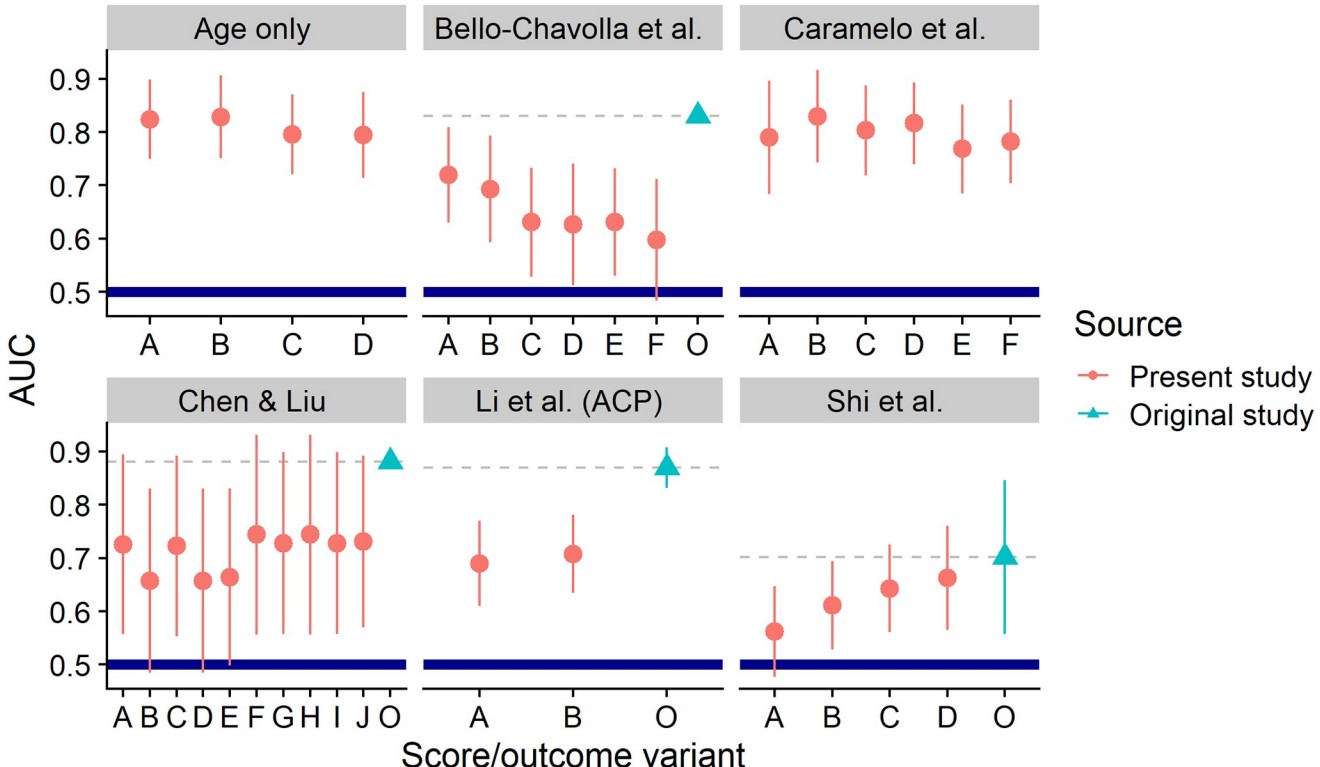

**Fig 6. Performance of tested prediction scores as measured by AUC.** AUC = 1 means perfect prediction while AUC ≤ 0.5 means that the score is worse than random guess and a better prediction would be obtained by reversing the score (marked by thick blue line). The line ranges represent the bootstrapped 95% confidence intervals. Red dots show results computed in present study—model variants (horizontal axis) vary in the outcome measured (12-day or 30-day mortality, severe disease) and potentially on how ambiguities in score computation were resolved, although this rarely makes a big difference—see S3 File for details. Cyan triangles show AUC as reported by the original authors or recomputed based on their published data. When the confidence interval or the AUC of the original study is not shown, it means that the value was not reported by the authors and not enough information to recompute it was given.

We do not believe our results should directly inform clinical decision, though we see some potential for inclusion of our results in future meta-analyses.

Regarding methodology, there are multiple approaches that are—at least in principle—capable of deriving causal conclusions from observational data, most notably the DAG framework [33, 34] and target trial framework [35, 36]. In all approaches—and also in some randomized designs—it is common that substantial uncertainty about the best statistical model for the task at hand remains and cannot be eliminated. Nevertheless, most published papers present results only from a single statistical model. We believe that this uncertainty about model choice should not be ignored, rather we should embrace the uncertainty and employ a multiverse analysis or other forms of robustness checks to explore how our conclusions would differ had different assumptions been adopted. In this work we tried to show how such an analysis could be performed and reported in practice. We note that modelling choices that are often made semi-arbitrarily, e.g., logistic regression for survival at 28 days vs. a Cox proportional hazards model for time to event, did in our case lead to substantially different results.

The hidden-Markov models (HMMs) used in this study are of some interest because they fitted the data well and allowed for inclusion of a larger number of predictors than the Cox proportional hazards model without making the posterior uncertainty unreasonably large. We believe this is because such models make better use of the available detailed data. Additionally, HMMs can be used even when the outcomes are observed only indirectly or noisily—as in the

original study that motivated our models which concerns the progression of Alzheimer's disease [20]. Noisily observed outcomes can pose problems for the proportional hazards model and require some special care [37, 38]. We should however note that the Markov property assumed in HMMs is likely to be a reasonable approximation in fewer settings than the proportional hazards assumption of the Cox model.

Common problems with prognostic models in medicine are small sample sizes used to develop the models, weak or problematic statistical methods and lack of external validation on independent datasets [39]. Those problems are prevalent also in prediction models for Covid-19 [14]. We used our dataset to validate several models and observed very poor performance for four out of the five models tested. In fact, the simple assumption that older patients tend to have worse outcomes provides better or similar results to all of the models we were able to implement. This is despite all of the scores including age as a predictor. There seem to be two causes—three of the models dichotomize the age into just two groups which is known to lose information [40, 41]. Of the other two models Chen & Liu [24] use stepwise variable selection which is known to be a problematic approch [42]. The resulting model puts largest relative weight on laboratory markers and deemphasizes age. Caramelo et al. [26] take the decade of age as a very strong predictor and perform the best on our data. Still their results are not distinguishable from just using age. Our findings are almost the same as in a similar but larger validation study using 22 models and 411 patients from the United Kingdom where no tested model provided better prediction for mortality than age alone [43].

## Conclusions

We provide very weak observational evidence against a substantial beneficiary effect of using azithromycin (both with or without hydroxychloroquine) and against substantial negative effect of using favipiravir in hospitalized Covid-19 patients. We also observed better outcomes associated with taking hydroxychloroquine, which is likely linked to substantial confounding by indication. Where our results contradict randomized trials, the most likely explanation is systematic bias in our dataset.

A lesson from our analysis is that the assessment of treatment efficacy from observational data is sensitive to modelling assumptions while it is usually almost impossible to determine which of the models is more likely to reflect reality (if any). We believe that using multiverse analysis is an appropriate way to explore data in such contexts as it lets us be transparent about this sensitivity. We further believe that using hidden Markov models is a promising complement to the standard Cox proportional hazards analysis when detailed information on disease progression is available, particularly because it lets us impose additional structure on the model and thus make inferences with more disease states than would be possible to handle in the Cox framework, making better use of the available data.

Additionally, our experience indicates that a substantial fraction of published prognostic models will perform much worse on new patients than on the datasets they were built for and that external validation is crucial. We suggest that comparing the prognostic models against simple baselines (e.g., decade of age as the single predictor) should be a first step in validation. Furthermore, some of the published scores lack enough information to let others implement the score in the same way.

## Supporting information

**S1 Fig. Data collection timeline.** Data collection periods at individual sites, showing the range of admission dates of patients included in the study. Note that we cannot provide additional information to link the sites here with data shown elsewhere as that would increase the risk of

deanonymization of the patients.
(TIFF)

**S2 Fig. Treatment combinations.** Upset plot of treatment combinations—each vertical bar displays the number of patients that received the combination indicated by filled dots in the matrix. Horizontal bars show the total number of patients receiveing the given treatment.
(TIFF)

**S3 Fig. Outcomes per site.** Number of patients and outcomes at the individual sites. The numbers above bars are the exact counts. Hospitalized = still hospitalized at the end of data collection at the site or transferred to other site and lost to followup. Sites are anonymized to preserve patient privacy.
(TIFF)

**S4 Fig. Markers and outcomes.** Density plots of worst marker values per patient, stratified by worst condition experienced by the patient. For each patient that had a given marker measured, the worst value was taken. Additionally the patients are classified by the worst condition (regardless of the timing relative to the worst marker levels). For each set of patients and marker an empirical density plot of the worst marker values is shown.
(TIFF)

**S5 Fig. Treatment onset.** Histogram of timing of first treatment relative to admission into one of the study sites. Two patients initiated treatment before admission, which is shown as the negative numbers.
(TIFF)

**S6 Fig. Association of HCQ with outcomes.** Estimates of model coefficients for association between hydroxychloroquine and main outcomes. The "Suspicious" section shows models that were found to not fit the data well or have computational issues—see supplementary statistical analysis for details. Each row represents a model—Categorical All/7/28 = Bayesian categorical regression for state at last observed day/day 7/day 28, Binary All/7/28 = Bayesian logistic regression for state at last observed day/day 7/day 28, Bayes Cox = Bayesian version of the Cox proportional hazards model with a binary outcome, Cox (single) = frequentist Cox model with a binary outcome, Cox (competing) = frequentist Cox model using competing risks (as in Fig 1a), HMM A = Bayesian hidden-Markov model as in Fig 1b with predictors for rate groups, HMM B = Bayesian hidden-markov model as in Fig 1b with predictors for individual rates, HMM C = Bayesian hidden-Markov model as in Fig 1c. For frequentist models, we show maximum likelihood estimate and 95% confidence interval, for Bayesian models we show posterior mean and 95% credible interval. The estimands are either log odds-ratio (Categorical, HMM) or log hazard ratio (Cox variants) or log ratio of mean duration of hospitalization (HMM duration). In all cases coefficient <0 means better patient outcome in the treatment group. Vertical lines indicate zero (blue) and substantial increase or decrease with odds or hazard ratio of 3:2 or 2:3 (green). Additionally the factors the model adjusted for are listed—Site = the study site, admitted = Admitted for Covid-19, Supportive = best supportive care initiated, Comorb. = total number of comorbidities, AZ = took azithromycin, HCQ = took hydroxychloroquine, FPV = took favipiravir, C. plasma = received convalescent plasma, first wave = only patients admitted before September 1st were included.
(TIFF)

**S7 Fig. Association of azithromycin with outcomes.** Estimates of model coefficients for association between azithromycin and main outcomes. The "Suspicious" section shows models that were found to not fit the data well or have computational issues—see supplementary

statistical analysis for details. Each row represents a model—Categorical All/7/28 = Bayesian categorical regression for state at last observed day/day 7/day 28, Binary All/7/28 = Bayesian logistic regression for state at last observed day/day 7/day 28, Bayes Cox = Bayesian version of the Cox proportional hazards model with a binary outcome, Cox (single) = frequentist Cox model with a binary outcome, Cox (competing) = frequentist Cox model using competing risks (as in Fig 1a), HMM A = Bayesian hidden-Markov model as in Fig 1b with predictors for rate groups, HMM B = Bayesian hidden-markov model as in Fig 1b with predictors for individual rates, HMM C = Bayesian hidden-Markov model as in Fig 1c. For frequentist models, we show maximum likelihood estimate and 95% confidence interval, for Bayesian models we show posterior mean and 95% credible interval. The estimands are either log odds-ratio (Categorical, HMM) or log hazard ratio (Cox variants) or log ratio of mean duration of hospitalization (HMM duration). In all cases coefficient <0 means better patient outcome in the treatment group. Vertical lines indicate zero (blue) and substantial increase or decrease with odds or hazard ratio of 3:2 or 2:3 (green). Additionally the factors the model adjusted for are listed—Site = the study site, admitted = Admitted for Covid-19, Supportive = best supportive care initiated, Comorb. = total number of comorbidities, AZ = took azithromycin, HCQ = took hydroxychloroquine, FPV = took favipiravir, C. plasma = received convalescent plasma, first wave = only patients admitted before September 1st were included.
(TIFF)

**S8 Fig. Association of favipiravir with outcomes.** Estimates of model coefficients for association between favipiravir and main outcomes. The "Suspicious" section shows models that were found to not fit the data well or have computational issues—see supplementary statistical analysis for details. Each row represents a model—Categorical All/7/28 = Bayesian categorical regression for state at last observed day/day 7/day 28, Binary All/7/28 = Bayesian logistic regression for state at last observed day/day 7/day 28, Bayes Cox = Bayesian version of the Cox proportional hazards model with a binary outcome, Cox (single) = frequentist Cox model with a binary outcome, Cox (competing) = frequentist Cox model using competing risks (as in Fig 1a), HMM A = Bayesian hidden-Markov model as in Fig 1b with predictors for rate groups, HMM B = Bayesian hidden-markov model as in Fig 1b with predictors for individual rates, HMM C = Bayesian hidden-Markov model as in Fig 1c. For frequentist models, we show maximum likelihood estimate and 95% confidence interval, for Bayesian models we show posterior mean and 95% credible interval. The estimands are either log odds-ratio (Categorical, HMM) or log hazard ratio (Cox variants) or log ratio of mean duration of hospitalization (HMM duration). In all cases coefficient <0 means better patient outcome in the treatment group. Vertical lines indicate zero (blue) and substantial increase or decrease with odds or hazard ratio of 3:2 or 2:3 (green). Additionally the factors the model adjusted for are listed—Site = the study site, admitted = Admitted for Covid-19, Supportive = best supportive care initiated, Comorb. = total number of comorbidities, AZ = took azithromycin, HCQ = took hydroxychloroquine, FPV = took favipiravir, C. plasma = received convalescent plasma, first wave = only patients admitted before September 1st were included.
(TIFF)

**S9 Fig. Association of markers with outcomes.** Estimates of model coefficients (log hazard ratios) for association between markers and death. The "Suspicious" section shows models that were found to not fit the data well or have computational issues, "Problematic" section shows models that were completely broken—see supplementary statistical analysis for details. Each row represents a model—Cox (competing) = frequentist Cox model using competing risks (as in Fig 1a), HMM A = Bayesian hidden-markov model as in Fig 1b with predictors for rate groups, JM = Bayesian joint longitudinal and time-to-event model. For frequentist

models, we show maximum likelihood estimate and 95% confidence interval, for Bayesian models we show posterior mean and 95% credible interval. Additionally the factors the model adjusted for are listed—Site = the study site, Supportive = best supportive care initiated, HCQ = took Hydroxychloroquine. We show posterior mean and 95% credible interval. (TIFF)

**S1 File. Data collection protocol.**
(PDF)

**S2 File. MS Excel table used for data collection.**
(XLSX)

**S3 File. Supplementary statistical analysis.** Contains details on all statistical models and procedures used.
(PDF)

## Author Contributions

**Conceptualization:** Martin Modrák, Daniel Suk, Adam Jaroš, Tomáš Hyánek.

**Data curation:** Martin Modrák, Tomáš Slisz, Grigorij Mesežnikov, Luis Fernando Casas-Mendez, Jaromír Vajter, Jan Táborský, Viktor Kubricht, Daniel Suk, Jan Horejsek, Martin Jedlička, Adriana Mifková, Adam Jaroš, Miroslav Kubiska, Jana Váchalová, Robin Šín, Markéta Veverková, Zbyšek Pospíšil, Julie Vohryzková, Rebeka Pokrievková, Kristián Hrušák, Kristína Christozova.

**Formal analysis:** Martin Modrák, Paul-Christian Bürkner, Tomáš Sieger, Vianey Leos-Barajas.

**Funding acquisition:** Martin Modrák.

**Investigation:** Martin Modrák, Paul-Christian Bürkner, Tomáš Sieger.

**Methodology:** Martin Modrák, Adam Jaroš, Vianey Leos-Barajas, Tomáš Hyánek.

**Project administration:** Martin Modrák.

**Resources:** Martin Modrák.

**Software:** Martin Modrák, Paul-Christian Bürkner, Tomáš Sieger.

**Supervision:** Martin Modrák, Martina Vašáková, Vianey Leos-Barajas, Karel Fišer, Tomáš Hyánek.

**Validation:** Martin Modrák.

**Visualization:** Martin Modrák.

**Writing – original draft:** Martin Modrák.

**Writing – review & editing:** Martin Modrák, Paul-Christian Bürkner, Tomáš Sieger, Tomáš Slisz, Martina Vašáková, Grigorij Mesežnikov, Luis Fernando Casas-Mendez, Jaromír Vajter, Jan Táborský, Viktor Kubricht, Daniel Suk, Jan Horejsek, Martin Jedlička, Adriana Mifková, Adam Jaroš, Miroslav Kubiska, Jana Váchalová, Robin Šín, Markéta Veverková, Zbyšek Pospíšil, Julie Vohryzková, Rebeka Pokrievková, Kristián Hrušák, Kristína Christozova, Vianey Leos-Barajas, Karel Fišer, Tomáš Hyánek.

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
