## [Decision Letter · Decision Letter 0]

3 Jun 2021

PONE-D-20-40463

Detailed disease progression of 213 patients hospitalized with Covid-19 in the Czech Republic: An exploratory analysis

PLOS ONE

Dear Dr. Modrak,

Thank you for submitting your manuscript to PLOS ONE. After careful consideration, we feel that it has merit but does not fully meet PLOS ONE’s publication criteria as it currently stands. Therefore, we invite you to submit a revised version of the manuscript that addresses the points raised during the review process.

We look forward to receiving your revised manuscript.

Kind regards,

Aleksandar R. Zivkovic

Academic Editor

PLOS ONE

2. In the ethics statement in the manuscript and in the online submission form, please provide additional information about the patient records used in your retrospective study, including:

a) whether all data were fully anonymized before you accessed them;

b) the date range (month and year) during which patients' medical records were accessed;

c) the date range (month and year) during which patients whose medical records were selected for this study sought treatment.

If patients provided informed written consent to have data from their medical records used in research, please include this information.

3. Please ensure you have thoroughly discussed any potential limitations of this study within the Discussion section, including the potential impact of confounding factors.

5. Please include captions for your Supporting Information files at the end of your manuscript, and update any in-text citations to match accordingly. Please see our Supporting Information guidelines for more information: http://journals.plos.org/plosone/s/supporting-information

Reviewers' comments:

Reviewer #1: The authors have provided an overview of COVID-19 hospitalized patients in the Czech Republic. The study is well-written in a proper English language and fashion, also, with quite dynamic scientific flow. The study cohort is rather small. However, there are few major concerns that should be clarified by the authors.

1. The "Introduction section" contains misleading information - Remdesivir is no longer available in only limited amounts of patients. Remdesivir is currently one of the front-line therapies.

2. The authors should explain why are the patients treated mostly with hydroxychloroquine despite only low clinical benefit (mentioned in the Introduction).

3. Table 1 should contain the total number of deaths. Also, the number of hydroxychloroquine treated patients is missing.

4. Given the presented data, it is clear that by the time of this study, the majority of patients was treated with either hydroxychloroquine or azithromycin. This has fundamentally changed during the past few months. The majority of patients is receiving dexamthasone and remdesivir, while in this study, the proportions are only 2%, and 1%, respectively.

This might imply two things:

either the included institutions have no access to first-line therapies or the included institutions are not following the treatment guidelines. Does this really reflect the treatment of COVID-19 patients in the Czech Republic in 2021? If so,

that is very disturbing.

At this point I believe that the authors have several options. They might collect a new up-to-date cohort and correlate the treatment strategies and outcomes of now and then. That would significantly improve the quality of the manuscript. In that case the manuscript title could reffer to "disease progression of patients hospitalized in the Czech R."

Other option is to highlight that these data are on the efficacy of hydroxychloroquine in the treatment of COVID-19 which was applied at the very beginning of the COVID pandemic. In such case, the manuscript title must change. Also the Introduction section should provide more data on hydroxychloroquine and the reasons for its application and also the study cohort should be presented as "remdesivir-naive" or dexamethasone-naive" which could be actually beneficial for the observation of hydroxychloroquine treatment of patients. Since Hydroxychloroquine is being used in a wide range of autoimmune diseases, the immune background of COVID-19 should be included. This study, however, cannot be titled as it currently stands. The title and the abstract must change.

5. The authors should at least discuss the current first-line treatments of COVID-19 in the Czech Republic and reffer to pertinent guidelines.

Reviewer #2: This study aimed at significant COVID-related health problems.

However, this study requires extensive revisions.

1. The organization of the text is complicated. The structure of this paper is far from a standard scientific article.

2. The sampling methods are unclear. The authors should clearly define sampling methods, research centers included in this study, characteristics of hospitals - single wards? or all hospitals were dedicated to COVID-19? Moreover, the timeline of the patients' enrolment of participants should be clearly defined.

3. More information about the COVID-19 pandemic in the Czech Republic is needed. Moreover, the Authors should present data on the organization of COVID-19 dedicated treatment.

4. Conclusions seem to be not supported by the results.

Reviewer #3: I like the general idea of this manuscript and find it interesting to the readers of Plos One. However, the conclusion section is very short and does not contain any references. I suggest rewriting the conclusion section and discussing your results in detail with reference to similar studies

---

## [Author Response · Author response to Decision Letter 0]

2 Jul 2021

The response to reviewers was uploaded as a separate file, but since the system requires us to fill this free text, we also put a copy here.

====== Reviewer #1 ======

Comment: The authors have provided an overview of COVID-19 hospitalized patients in the Czech Republic. The study is well-written in a proper English language and fashion, also, with quite dynamic scientific flow. The study cohort is rather small. However, there are few major concerns that should be clarified by the authors.

Response: Thanks for the kind words. We believe a potential source of confusion is that the paper has unfortunately spent substantial time between submission to PLoS One and being sent out for review. The data were collected in the early phases of the pandemic (March - October 2020) and we submitted the manuscript in December 2020. Due to this delay (which is not the reviewers fault) the information has necessarily become outdated. We rewrote the Introduction section to clarify the timeline and make a better distinction between the state of knowledge at the time the patients were hospitalized and current state of knowledge.

Comment: 1. The "Introduction section" contains misleading information - Remdesivir is no longer available in only limited amounts of patients. Remdesivir is currently one of the front-line therapies.

Response: We updated the introduction with up-to-date information on Covid treatments and contrasted them with what was available upon data collection.

Comment: 2. The authors should explain why are the patients treated mostly with hydroxychloroquine despite only low clinical benefit (mentioned in the Introduction).

Response: We clarified the timeline and discuss what information was available at the time for which we collected data.

Comment: 3. Table 1 should contain the total number of deaths. Also, the number of hydroxychloroquine treated patients is missing.

Response: We changed “Deceased” to “Death” in Table 1 to avoid any confusion. We also duplicated the information on hydroxychloroquine treatment to be shown in both the columns of Table 1 as well as in rows to keep all of the drug information in a consistent format.

Comment: 4. Given the presented data, it is clear that by the time of this study, the majority of patients was treated with either hydroxychloroquine or azithromycin. This has fundamentally changed during the past few months. The majority of patients is receiving dexamthasone and remdesivir, while in this study, the proportions are only 2%, and 1%, respectively.

This might imply two things:

either the included institutions have no access to first-line therapies or the included institutions are not following the treatment guidelines. Does this really reflect the treatment of COVID-19 patients in the Czech Republic in 2021? If so,

that is very disturbing.

Response: The data reflect the state in Czech Republic at the early stages of the pandemic when the choice of treatments was different. We clarified this in the Introduction section. 

Comment: At this point I believe that the authors have several options. They might collect a new up-to-date cohort and correlate the treatment strategies and outcomes of now and then. That would significantly improve the quality of the manuscript. In that case the manuscript title could reffer to "disease progression of patients hospitalized in the Czech R."

Other option is to highlight that these data are on the efficacy of hydroxychloroquine in the treatment of COVID-19 which was applied at the very beginning of the COVID pandemic. In such case, the manuscript title must change. 

Response: Unfortunately collecting new data is not feasible. We changed the title to reflect that the data correspond to early phases of the pandemic. As we indicated in the paper, we believe that given the progress made in understanding the efficacy of hydroxychloroquine in the past year, our results on hydroxychloroquine are not of primary interest (although we report them as this was the initial primary aim of the study). We believe the currently more relevant results are our evaluation of the published prognostic models and our data on favipiravir, which despite the low number of patients are of some interest as favipiravir is comparatively underreported in the literature (we were already contacted for inclusion of the data in meta analyses).

Comment: Also the Introduction section should provide more data on hydroxychloroquine and the reasons for its application and also the study cohort should be presented as "remdesivir-naive" or dexamethasone-naive" which could be actually beneficial for the observation of hydroxychloroquine treatment of patients. Since Hydroxychloroquine is being used in a wide range of autoimmune diseases, the immune background of COVID-19 should be included. This study, however, cannot be titled as it currently stands. The title and the abstract must change.

Response: We clarified the treatment choices and timeline and made it clear in title and abstract that the data are from early phases of the pandemic when remdesivir and dexamethasone were not routinely used.

Comment: 5. The authors should at least discuss the current first-line treatments of COVID-19 in the Czech Republic and reffer to pertinent guidelines.

 Response: References to the current guidelines were added into the Discussion section.

====== Reviewer #2 ======

Comment: 1. The organization of the text is complicated. The structure of this paper is far from a standard scientific article.

Response: We agree that since we discuss multiple results derived from the same dataset, the structure of the paper is more complex as each of the results needs to be discussed in methods, results and conclusions. We are however unsure how this could be improved without removing content from the paper. We made minor changes to headings to more closely match similar works already published in PLoS One (e.g. https://journals.plos.org/plosone/article?id=10.1371/journal.pone.0249346) and added a Discussion section.

Comment: 2. The sampling methods are unclear. The authors should clearly define sampling methods, research centers included in this study, characteristics of hospitals - single wards? or all hospitals were dedicated to COVID-19? Moreover, the timeline of the patients' enrolment of participants should be clearly defined.

Response: We clarified the sampling methods and timeline in the “Patients and study design” subsection. We note that the full timeline for each site was given in Supplementary Figure 1. 

Comment: 3. More information about the COVID-19 pandemic in the Czech Republic is needed. Moreover, the Authors should present data on the organization of COVID-19 dedicated treatment.

Response: Information on the pandemic in Czech Republic was added in the Introduction section.

Comment: 4. Conclusions seem to be not supported by the results.

Response: We are not sure which conclusions the reviewer assumes to be unsupported. We added a Discussion section to make the connection between Results and Conclusion sections more explicit.

====== Reviewer #3 ======

Comment:I like the general idea of this manuscript and find it interesting to the readers of Plos One. However, the conclusion section is very short and does not contain any references. I suggest rewriting the conclusion section and discussing your results in detail with reference to similar studies

Response: Thanks for the kind words. We added a Discussion section that discusses the results in more details and adds references.

---

## [Editor Report · Decision Letter 1]

16 Jul 2021

Disease progression of 213 patients hospitalized with Covid-19 in the Czech Republic in March - October 2020: An exploratory analysis

PONE-D-20-40463R1

Dear Dr. Modrak,

We’re pleased to inform you that your manuscript has been judged scientifically suitable for publication and will be formally accepted for publication once it meets all outstanding technical requirements.

Kind regards,

Aleksandar R. Zivkovic

Academic Editor

PLOS ONE

---

## [Editor Report · Acceptance letter]

7 Sep 2021

PONE-D-20-40463R1 

Disease progression of 213 patients hospitalized with Covid-19 in the Czech Republic in March - October 2020: An exploratory analysis 

Dear Dr. Modrák:

I'm pleased to inform you that your manuscript has been deemed suitable for publication in PLOS ONE. Congratulations! Your manuscript is now with our production department. 

Kind regards, 

on behalf of

Dr. Aleksandar R. Zivkovic 

Academic Editor

PLOS ONE